# MULTI-TASK MULTICRITERIA HYPERPARAMETER OPTIMIZATION

## ABSTRACT

We present a new method for searching optimal hyperparameters among several tasks and several criteria. Multi-Task Multi Criteria method (MTMC) provides several Pareto-optimal solutions, among which one solution is selected with given criteria significance coefficients. The article begins with a mathematical formulation of the problem of choosing optimal hyperparameters. Then, the steps of the MTMC method that solves this problem are described. The proposed method is evaluated on the image classification problem using a convolutional neural network. The article presents optimal hyperparameters for various criteria significance coefficients.

## 1 INTRODUCTION

Hyperparameter optimization (Hutter et al., 2009) is an important component in the implementation of machine learning models (for example, logistic regression, neural networks, SVM, gradient boosting, etc.) in solving various tasks, such as classification, regression, ranking, etc. The problem is how to choose the optimal parameters when a trained model is evaluated using several sets and several criteria.

This article describes a method to solving the above problem. We will present the results of experiments on the selection of hyperparameters obtained using the proposed approach (MTMC) with various criteria significance coefficients.

The article is organized as follows. First, we discuss related work in Section 2. Section 3 describes the proposed method. Section 4 presents the results of experiments on the selection of optimal hyperparameters. Section 5 contains the conclusion and future work.

## 2 RELATED WORK

Hyperparameter optimization is applied to solve various problems such as computer vision (Bergstra et al., 2013; Dong et al., 2019), robotics (Mahmood et al., 2018; Tran et al., 2020), natural language processing (Wang et al., 2015; Dernoncourt & Lee, 2016) and speech synthesis (Koriyama et al., 2014).

The problem of choosing optimal hyperparameters has long been known. Existing solutions can be considered for the following features:

1. Number of optimal solutions.

2. Number of tasks to be solved.

3. Number of criteria for choosing the optimal solution.

In this article, *task* means a set of images with a number of classes $N_{classes}$ and a number of images $N_{images}$. There are examples of the same classes between tasks, the difference is how the images are made (different lighting, background and used cameras). *Criteria* is a quantitative characteristic of training / evaluation a neural network on a task (e.g. accuracy, latency or epoch of training convergence).

In (Sener & Koltun, 2018), a Pareto optimization method is proposed, in which the optimal solution is given for several problems simultaneously. This method consists in minimizing the weighted sum of loss functions for each task. (Fliege & Svaiter, 2000) describes the Pareto optimization method, which gives an optimal solution according to several criteria based on gradient descent, and this optimization is also carried out in the learning process. In (Igel, 2005), the search for a Pareto-optimal solution is carried out according to several criteria. (Miettinen, 2012) gives several methods for multiobjective Pareto-optimization. The method in (Bengio, 2000) gives optimal hyperparameters using back propagation through the Cholesky decomposition. In (Bergstra et al., 2011), optimization is performed using a random choice of hyperparameters based on the expected improvement criterion. (Bergstra & Bengio, 2012) proposes method of hyperparametric optimization based on random search in the space of hyperparameters. In (Snoek et al., 2012), search for optimal hyperparameters is carried out using Bayesian optimization. (Swersky et al., 2013) proposes a method for finding optimal hyperparameters using multi-task Bayesian optimization. (Paria et al., 2020; Hernández-Lobato et al., 2016) describe the Bayesian methods for multi-objective optimization.

The proposed method based on Pareto optimization. Pareto optimality means that it is impossible to improve the Pareto optimal solution by any criteria without worsening it by at least one other criteria. Thus, for a certain set of values of the criteria, the Pareto-selected solutions are optimal. The closest criteria to the given criteria are determined by finding the minimum weighted sum of Pareto solutions (where the weight is the inverse criteria).

The novelty of MTMC method:

1. Optimization is carried out simultaneously according to several criteria and several tasks with setting the significance of the criteria.

2. The choice of optimal hyperparameters is provided *after* training and evaluation, which eliminates the need to re-train the model.

3. The proposed method does not need to be trained.

## 3 THE PROPOSED METHOD

First, we describe the mathematical problem that MTMC solves, then we present the steps performed in MTMC.

### 3.1 FORMALIZATION OF THE PROBLEM

In the proposed method, the model is evaluated on several test sets (tasks) $T$. The problem of finding a minimum for tasks $T$ is known as minimizing the expected value of empirical risk (Vapnik, 1992).

The choosing optimal hyperparameters is formalized as follows:

$$\theta = \operatorname*{argmin}_{\theta \in \Theta} \mathbb{E}_\tau [\mathcal{L}(\theta, \phi)] \tag{1}$$

where $\Theta$ is the set of all hyperparameters, $\theta$ is the selected optimal hyperparameters, $\phi$ is the vector of significance coefficients of the criteria, $\mathcal{L}(\cdot)$ is the estimation function of the model with the given hyperparameters $\theta$ and the coefficients $\phi$, $\tau$ is the task for which optimization is performed.

The developed method gives a solution to the problem (1).

### 3.2 DESCRIPTION OF MTMC

According to (1), the developed method should fulfill the following requirements:

1) the method should solve the minimization problem;

2) the significance of each criterion is determined by the vector of coefficients $\phi$ (the higher the coefficient, the more important the corresponding criterion).

We denote the test sample of the task $\tau$:

$$x^i \sim \mathcal{D}, i = 1 \dots N_{\text{task}} \tag{2}$$

where $x^i$ is the $i^{th}$ test set has the distribution $\mathcal{D}$, $N_{\text{task}}$ is the number of tasks.

Before choosing hyperparameters, for model $\mathcal{M}$ we obtain an evaluation matrix for the test set $x^i$ and the given evaluation criteria:

$$V = \mathcal{M}\left(x^i; \Theta\right) \tag{3}$$

$$\mathcal{M}\left(x^i; \Theta\right) :: \left(\mathbb{R}^{x_{\text{size}}}, \mathbb{R}^{N_{\text{parameter}} \times N_{\text{combination}}}\right) \to \mathbb{R}^{N_{\text{combination}} \times N_{\text{criteria}}} \tag{4}$$

where $\mathcal{M}(\cdot)$ is the model function that transforms the given set $x^i$ and with the given hyperparameters $\Theta$ into the evaluation matrix $V$, $N_{\text{criteria}}$ is the number of criteria, $x_{\text{size}}$ is the dimension of the test set, $N_{\text{parameter}}$ is the number of hyperparameters, $N_{\text{combination}}$ is the number of hyperparameter combinations.

Then, the function $\mathcal{L}$ is calculated for each set $x^i$, which is formally described as follows:

$$\mathcal{L}(\cdot; \Theta, \phi) = \mathcal{E}(V; \phi) \tag{5}$$

$$\mathcal{E}(V; \phi) :: \left(\mathbb{R}^{N_{\text{criteria}}}, \mathbb{R}^{N_{\text{criteria}}}\right) \to \mathbb{R}^1 \tag{6}$$

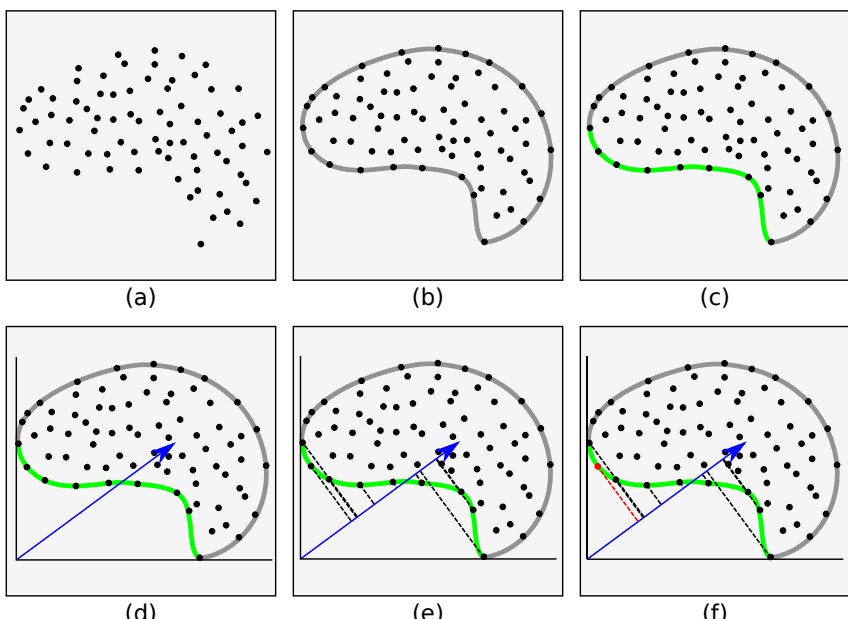

Figure 1: Illustration of the steps of our proposed method MTMC. (a) Evaluation for all combinations of hyperparameters in the criteria space. (b) Getting the evaluation boundary. (c) Getting Pareto optimal solutions. (d) Defining the criteria vector. (e) Projecting Pareto optimal solutions onto the criteria vector. (f) Finding the closest projection to the origin of coordinates.

Figure 1 shows the steps of MTMC to obtain the optimal solution for the given criteria. MTMC method gives Pareto optimal solutions in which the following steps are performed:

1. The vectors from the evaluation $V$ (the number of such vectors is $N_{\text{criteria}}$) is in the space of given criteria.

2. Pareto gives solutions that are closer to optimal by some criterion, and farther from optimal by other criteria. We get Pareto optimal solutions $\tilde{V} \subseteq V$ the nearest Pareto front to the origin of the criteria space coordinates:

$$\tilde{V} = \text{ParetoFront}(V), \quad \tilde{V} \in \mathbb{R}^{N_{\text{opt}} \times N_{\text{criteria}}} \tag{7}$$

where $N_{\text{opt}}$ is the number of Pareto optimal solutions.

3. The optimal solutions $\tilde{v} \in \tilde{V}$ are scaled according to each criterion to the interval $[0; 1]$:

$$\tilde{V}_{\text{scaled}} = \frac{\tilde{V}_i - \tilde{v}_{\text{min}}}{\tilde{v}_{\text{max}} - \tilde{v}_{\text{min}}}, \tilde{v}_{\text{min}} \in \mathbb{R}^{N_{\text{criteria}}}, \tilde{v}_{\text{min}} \in \mathbb{R}^{N_{\text{criteria}}}, i = 1 \ldots N_{opt} \tag{8}$$

where $\tilde{v}_{max}$ is the vector of maximum values of $\tilde{V}$ for each criterion, $\tilde{v}_{min}$ is the vector of minimum values of $\tilde{V}$ for each criterion.

Thus, the optimal solution is the solution closest to the origin of coordinates, and if any solution $\tilde{v} \in \tilde{V}$ is the origin of coordinates, then it is optimal for any $\phi$.

4. The vector $\phi$ in the space of criteria is defined.

We introduce the vector of the optimal solution, which is the middle of the segment $[0; 1]$ in the axes of the criteria space:

$$\phi_{opt} = \left( \forall i : \phi_0 = \cdots = \phi_i = \cdots = \phi_{N_{criteria}} = \frac{1}{2} \right). \tag{9}$$

Conditions for $\phi$ are:

$$\phi = \begin{cases} \phi_{\text{opt}}, & \text{if } \forall i : \phi_i = 0, \\ \phi \in [0; 1], & \text{otherwise.} \end{cases} \tag{10}$$

5. It is necessary to determine which hyperparameter in the criteria space is closer to the given criterion. Project the vectors from the matrix $\tilde{V}_{scaled}$ onto the vector $\phi$:

$$\tilde{V}_{proj} = \frac{\tilde{V}_{scaled}^T \cdot \phi}{\|\phi\|}, \quad \tilde{V} \in \mathbb{R}^{N_{opt}}. \tag{11}$$

From (9) and (11) it follows that if the vectors $\phi$ and $\phi_{opt}$ are collinear, then:

$$\exists \lambda : \phi = \lambda \cdot \phi_{opt} \Rightarrow \tilde{V}_{proj} = \sum_i \left[ \frac{1}{\phi_i} \cdot \frac{\tilde{V}_{scaled_i}}{\|\phi\|} \right] \propto \sum_i \tilde{V}_{scaled_i} = \left\| \tilde{V}_{scaled} \right\|_1. \tag{12}$$

That is, in the case of equality of all elements of $\phi$, the minimization problem reduces to finding the minimum $L1$-norm $\tilde{V}_{scaled}$.

From (11) it also follows that if some component of the vector $\phi$ is equal to zero, then the corresponding criterion will not affect the choice of the optimal hyperparameter. If all criteria are equal to zero, except for one, then only the criterion with a nonzero component of the vector $\phi$ will affect the choice of optimal hyperparameters.

6. We find hyperparameters $\theta$ at which the minimum of the vector $\tilde{V}_{proj}$ is reached, which is equivalent to finding the minimum weighted sum of vector values:

$$\theta = \underset{\theta}{\text{argmin}} \tilde{V}_{\text{proj}}. \tag{13}$$

Figure 2 shows an example solution using MTMC for random numbers in the three-dimensional space.

## 4 CONDUCTING EXPERIMENTS

First, the evaluation matrix $V$ for the selected model $\mathcal{M}$ is obtained. Then, for various combinations of components of $\phi$, optimal hyperparameters are selected using MTMC.

### 4.1 OBTAINING THE EVALUATION MATRIX

The developed MTMC method is applied to solve the problem of image classification. The problem we are solving is described in the article (Akhmetzyanov & Yuzhakov, 2019). The problem is in the choice of such hyperparameters, which achieve the highest classification accuracy among several tasks. Each task consists of several images of one of two objects such as a plastic bottle and other object. The tasks differ in how the images are made, namely lighting, background and used cameras.

In (Akhmetzyanov & Yuzhakov, 2018), we selected the MobileNet neural network architecture (Howard et al., 2017) as a mathematical model for image processing.

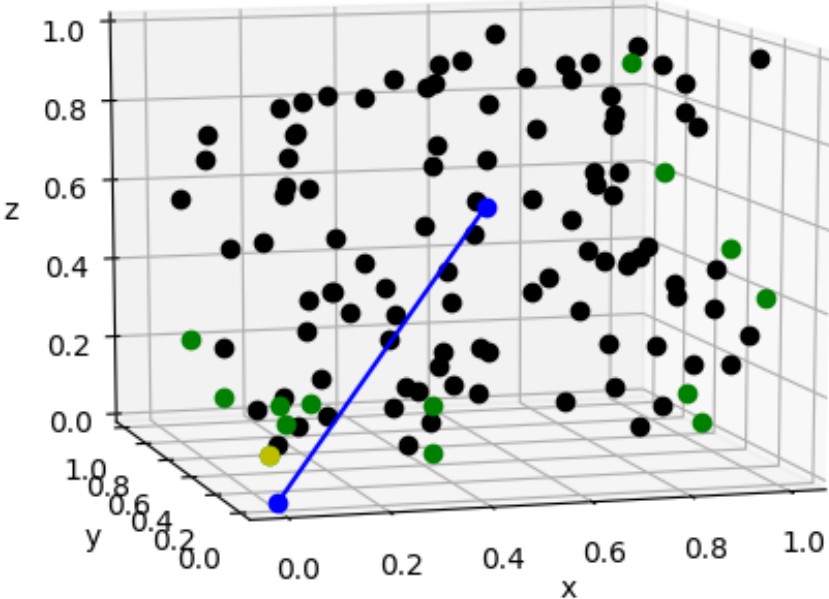

Figure 2: Example of a solution given by MTMC, green points denote Pareto optimal solutions, blue vector is the vector $\phi$, yellow point denotes the optimal solution given by MTMC for a given $\phi$.

The search for optimal hyperparameters was carried out among two popular training methods: changing the learning speed based on the epoch $lr = base\_lr \cdot lr\_decay^{epoch}$ (where $base\_lr$ is the initial learning rate, $lr\_decay$ is the coefficient of decreasing learning rate, $epoch$ is the number of epochs) and cyclical learning (Smith, 2017). In cyclic learning, there are three ways to change the learning rate:

1. *triangular* is fixed initial learning rate (*base_lr*), maximum fixed learning rate (*max_lr*), learning rate increases from *base_lr* to *max_lr* and decreases from *max_lr* to *base_lr* linearly.

2. *triangular2* is fixed initial learning rate (*base_lr*), maximum learning rate (*max_lr*), learning rate, as in triangular, varies linearly, but *max_lr* in the learning process is halved.

3. *exp_range* is fixed initial learning rate (*base_lr*), maximum learning rate (*max_lr*), learning rate also changes linearly, but *max_lr* in the learning process decreases exponentially.

In the first learning method, the hyperparameters are the value of the initial learning rate (*base_lr*) and the coefficient of decreasing learning rate (*lr_decay*). In the second method, hyperparameters a way to change the learning rate (*cyclic_mode*), the value of the initial learning rate (*base_lr*) and maximum learning rate (*max_lr*).

For each hyperparameter, a range of change and a constant step of change within the range were selected. For training, Grid search was used among $N_{combination} = 100$ combinations of hyperparameters.

For each combination of hyperparameters, training was carried out using cross-validation k-fold (Stone, 1974) with 10 folds. For training, Keras (Chollet et al., 2015) and TensorFlow (Abadi et al., 2015) were used. The training lasted 15 epochs; the test was carried out on $N_{task} = 5$ different test sets. That is, $100 \cdot 10 = 1000$ is number of different neural networks, $15 \cdot 1000 = 15000$ neural networks evaluations are conducted, $15000 \cdot 5 = 75000$ evaluation results are obtained. Neural networks trained on ten TPUs v2, which took several days.

Among all epochs, for each fold and for each test set, the maximum accuracy is selected, as well as the number of the epoch at which the maximum accuracy is achieved. The following values are calculated for each test set among the folds: the expected value and the variance of the classification error, the expected value and the variance of the epoch number at which convergence on the

test set is achieved. We have obtained an evaluation matrix among all neural networks with their hyperparameters and among all test samples.

## 4.2 PROCESSING THE EVALUATION MATRIX

Based on (1), for each criterion, among all the samples, the expected value is considered. That is, for all test sets, the criteria: (i) the sample mean of the classification error, (ii) the sample variance of the classification error, (iii) the sample mean and (iv) sample variance of the epoch number at which convergence on the test set is achieved. These values are the criteria for evaluating hyperparameters for a certain test set (matrix $V$ from (3)) with the number of criteria $N_{criteria} = 4$.

$\tilde{V}$ is calculated from (7), the number of Pareto optimal solutions obtained is $N_{opt} = 25$. Optimal hyperparameters, i.e., $\tilde{V}$, are presented in Appendix A.

The vector of the optimal solution according to (9) for $N_{criteria} = 4$ is $\phi_{opt} = \{0.5; 0.5; 0.5; 0.5\}$. Next, calculations are carried out according to (8) and (11), and for various $\phi$ optimal solutions are chosen according to (5). These optimal solutions are presented in Appendix B. Also, Appendix C shows the learning curve of the neural network for each test set and each epoch for the obtained optimal hyperparameters.

## 5 CONCLUSION

In this work, we proposed a new method for hyperparameter optimization among several tasks and several criteria. We trained several neural networks with various hyperparameters to solve the image classification problem. Then, for these neural networks, evaluation matrices were obtained on several tasks. We applied MTMC to these matrices and got optimal solutions with different significance coefficients. In the future, we will work to create a meta-learning method that solves the same problem as the method described in this article, but optimization will be performed among various models.

### ACKNOWLEDGMENTS

The reported study was partially supported by the Government of Perm Krai, research project No. C-26/174.6.

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

Table 1: Pareto optimal solutions for the first learning method

| base_lr | lr_decay |
|---------|----------|
| 0.001   | 0.75     |
| 0.001   | 0.8      |
| 0.005   | 0.75     |
| 0.01    | 0.9      |
| 0.01    | 0.95     |

Table 2: Pareto optimal solutions for the second learning method

| base_lr | max_lr | cyclic_mode |
|---------|--------|-------------|
| 0.0001  | 0.005  | exp_range   |
| 0.0001  | 0.005  | triangular2 |
| 0.0005  | 0.001  | exp_range   |
| 0.0005  | 0.005  | triangular2 |
| 0.001   | 0.0001 | triangular2 |
| 0.001   | 0.0005 | triangular  |
| 0.001   | 0.001  | exp_range   |
| 0.001   | 0.005  | triangular2 |
| 0.005   | 0.0001 | triangular  |
| 0.005   | 0.005  | triangular  |
| 0.01    | 0.0001 | triangular  |
| 0.01    | 0.0001 | triangular2 |
| 0.01    | 0.005  | triangular  |
| 0.01    | 0.005  | triangular2 |
| 0.01    | 0.01   | triangular  |
| 0.0001  | 0.0001 | triangular  |
| 0.0005  | 0.001  | triangular  |
| 0.0005  | 0.01   | triangular2 |
| 0.001   | 0.005  | triangular  |
| 0.005   | 0.01   | triangular  |

## A  PARETO OPTIMAL HYPERPARAMETERS

Table 1 and Table 2 show the Pareto optimal hyperparameters for the two learning methods.

## B  MTMC Optimal Hyperparameters

Table 3 shows the optimal hyperparameters $\theta$ obtained using MTMC method for given significance coefficients of the criteria $\phi$.

## C  Accuracy of MTMC Optimal Solutions

The figures 3-7 show optimal solutions chosen by MTMC method: 95% confidence intervals of the dependence of accuracy on the fold / epoch and maximum accuracy for each of the folds (**left**) and box plot with maximum accuracy for all test sets (**right**).

Table 3: Optimal hyperparameters for relevant criteria significance coefficients

| $\phi_0$ | $\phi_1$ | $\phi_2$ | $\phi_3$ | $\theta$ |
|---|---|---|---|---|
| 0.5 | 0.5 | 0.5 | 0.5 | base_lr=0.01, max_lr=0.01, cyclic_mode=triangular |
| 0.0 | 0.5 | 0.5 | 0.5 | base_lr=0.01, max_lr=0.01, cyclic_mode=triangular |
| 1.0 | 0.5 | 0.5 | 0.5 | base_lr=0.01, max_lr=0.01, cyclic_mode=triangular |
| 0.5 | 0.0 | 0.5 | 0.5 | base_lr=0.005, max_lr=0.0001, cyclic_mode=triangular |
| 0.5 | 1.0 | 0.5 | 0.5 | base_lr=0.01, max_lr=0.01, cyclic_mode=triangular |
| 0.5 | 0.5 | 0.0 | 0.5 | base_lr=0.01, max_lr=0.01, cyclic_mode=triangular |
| 0.5 | 0.5 | 1.0 | 0.5 | base_lr=0.01, max_lr=0.01, cyclic_mode=triangular |
| 0.5 | 0.5 | 0.5 | 0.0 | base_lr=0.0001, max_lr=0.005, cyclic_mode=triangular2 |
| 0.5 | 0.5 | 0.5 | 1.0 | base_lr=0.01, max_lr=0.01, cyclic_mode=triangular |
| 0.0 | 0.0 | 0.5 | 0.5 | max_lr=0.005, lr_decay=0.75 |
| 1.0 | 1.0 | 0.5 | 0.5 | base_lr=0.01, max_lr=0.01, cyclic_mode=triangular |
| 0.5 | 0.5 | 0.0 | 0.0 | base_lr=0.0001, max_lr=0.005, cyclic_mode=triangular2 |
| 0.5 | 0.5 | 1.0 | 1.0 | base_lr=0.01, max_lr=0.01, cyclic_mode=triangular |
| 1.0 | 0.0 | 0.0 | 0.0 | base_lr=0.0001, max_lr=0.005, cyclic_mode=triangular2 |
| 0.0 | 1.0 | 0.0 | 0.0 | base_lr=0.0001, max_lr=0.005, cyclic_mode=triangular2 |
| 0.0 | 0.0 | 1.0 | 0.0 | max_lr=0.005, lr_decay=0.75 |
| 0.0 | 0.0 | 0.0 | 1.0 | base_lr=0.0005, max_lr=0.001, cyclic_mode=exp_range |

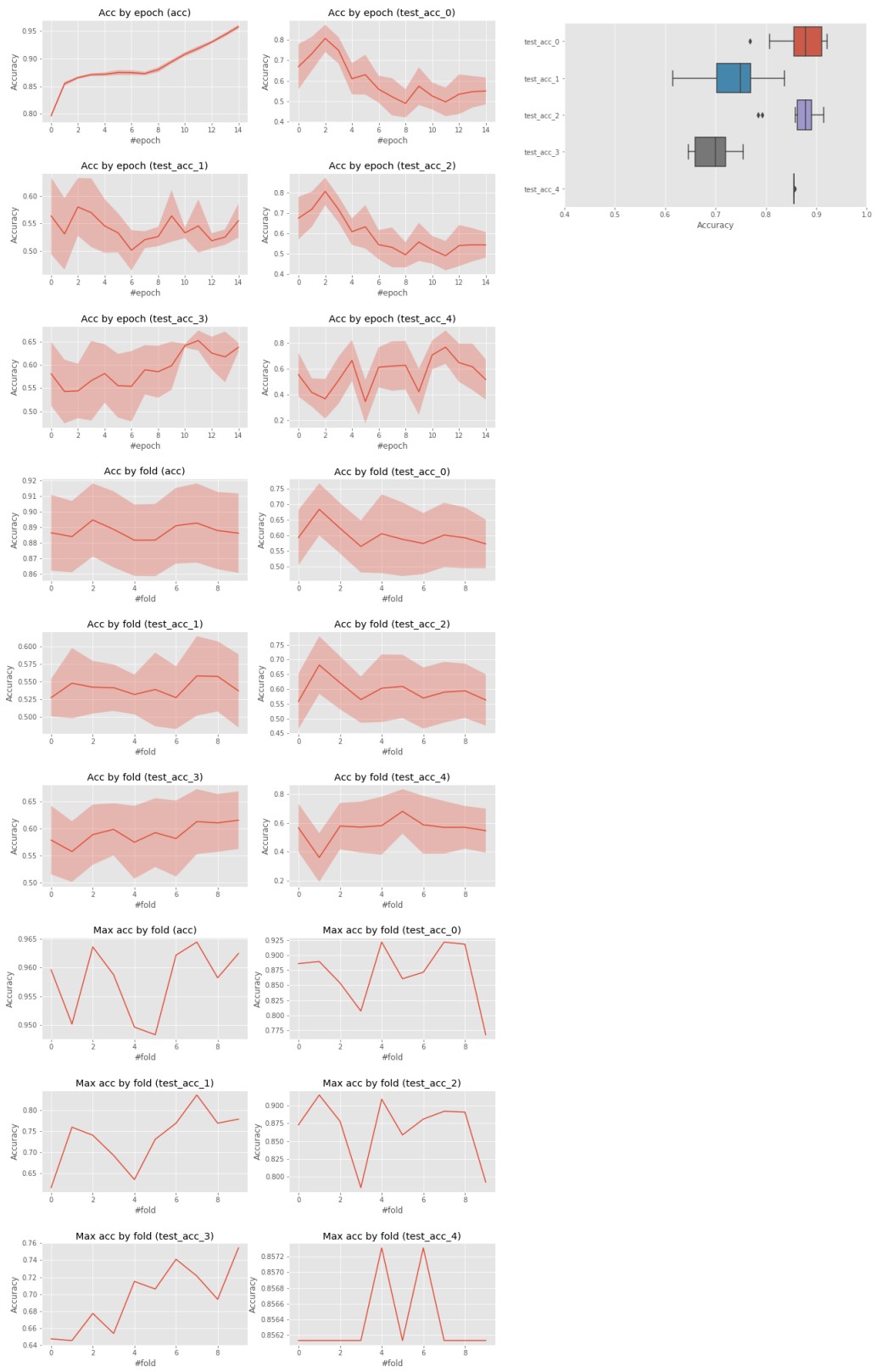

Figure 3: base_lr=0.01, max_lr=0.01, cyclic_mode=triangular

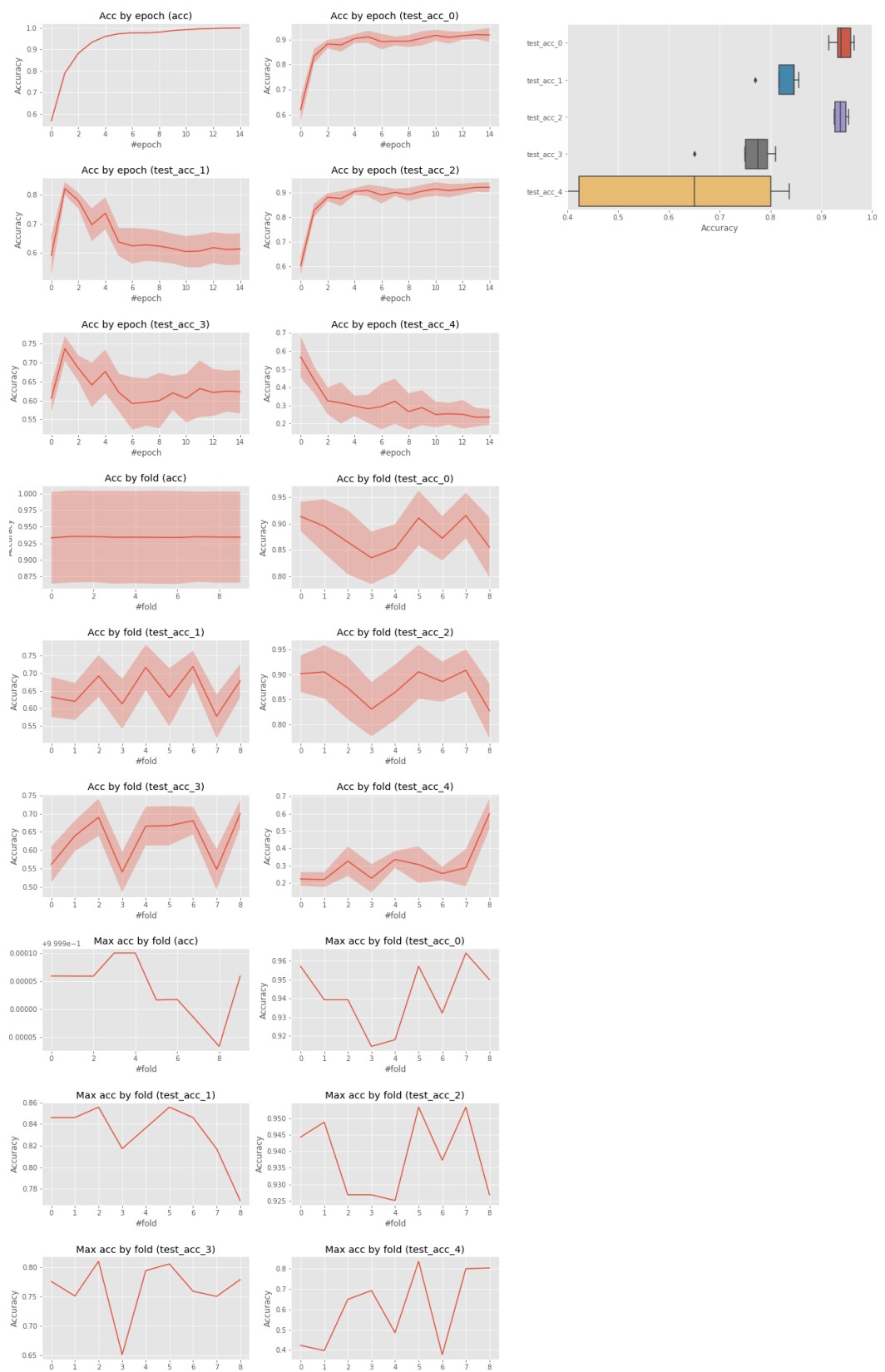

Figure 4: base_lr=0.005, max_lr=0.0001, cyclic_mode=triangular

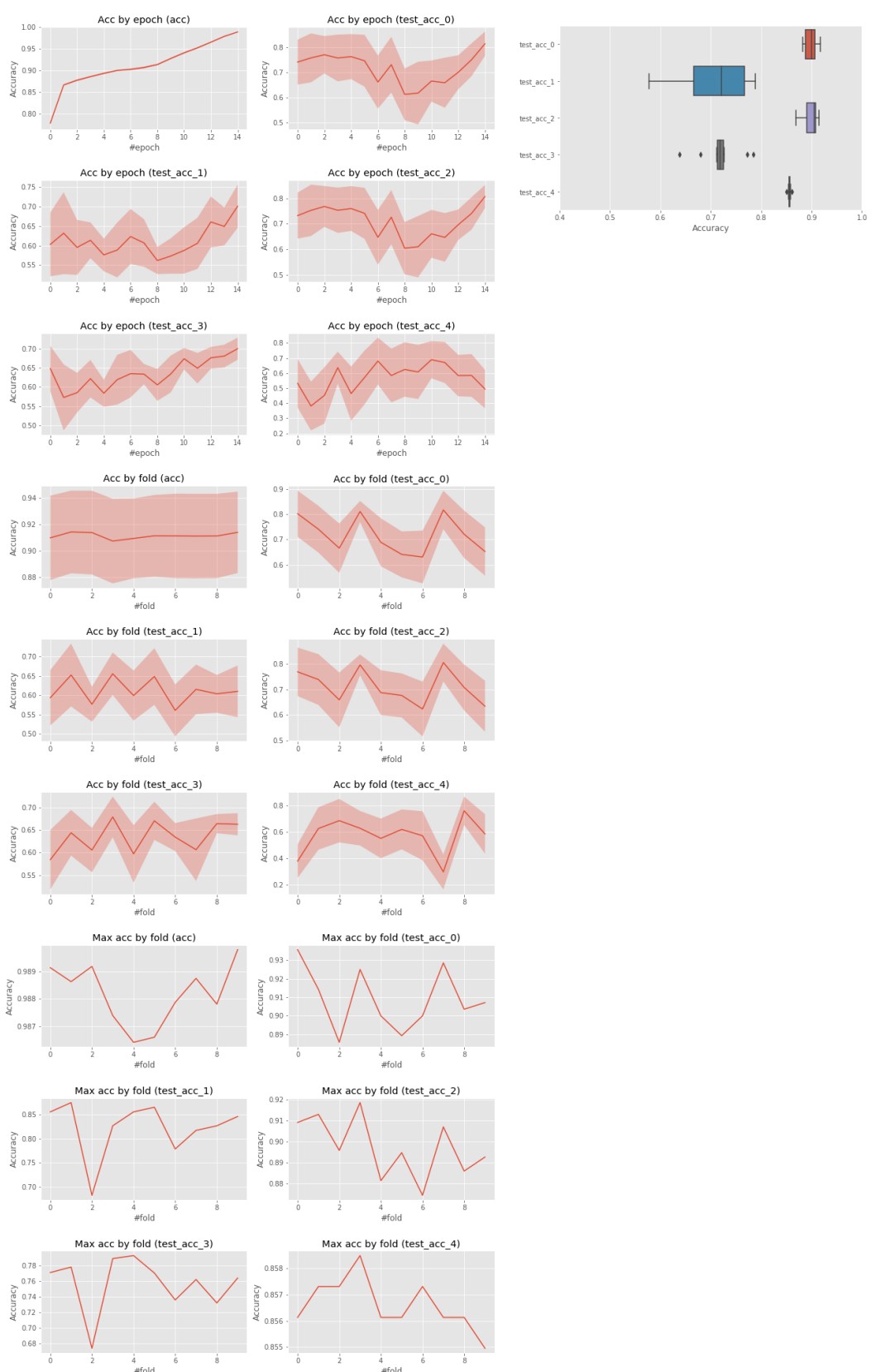

Figure 5: base_lr=0.0001, max_lr=0.005, cyclic_mode=triangular2

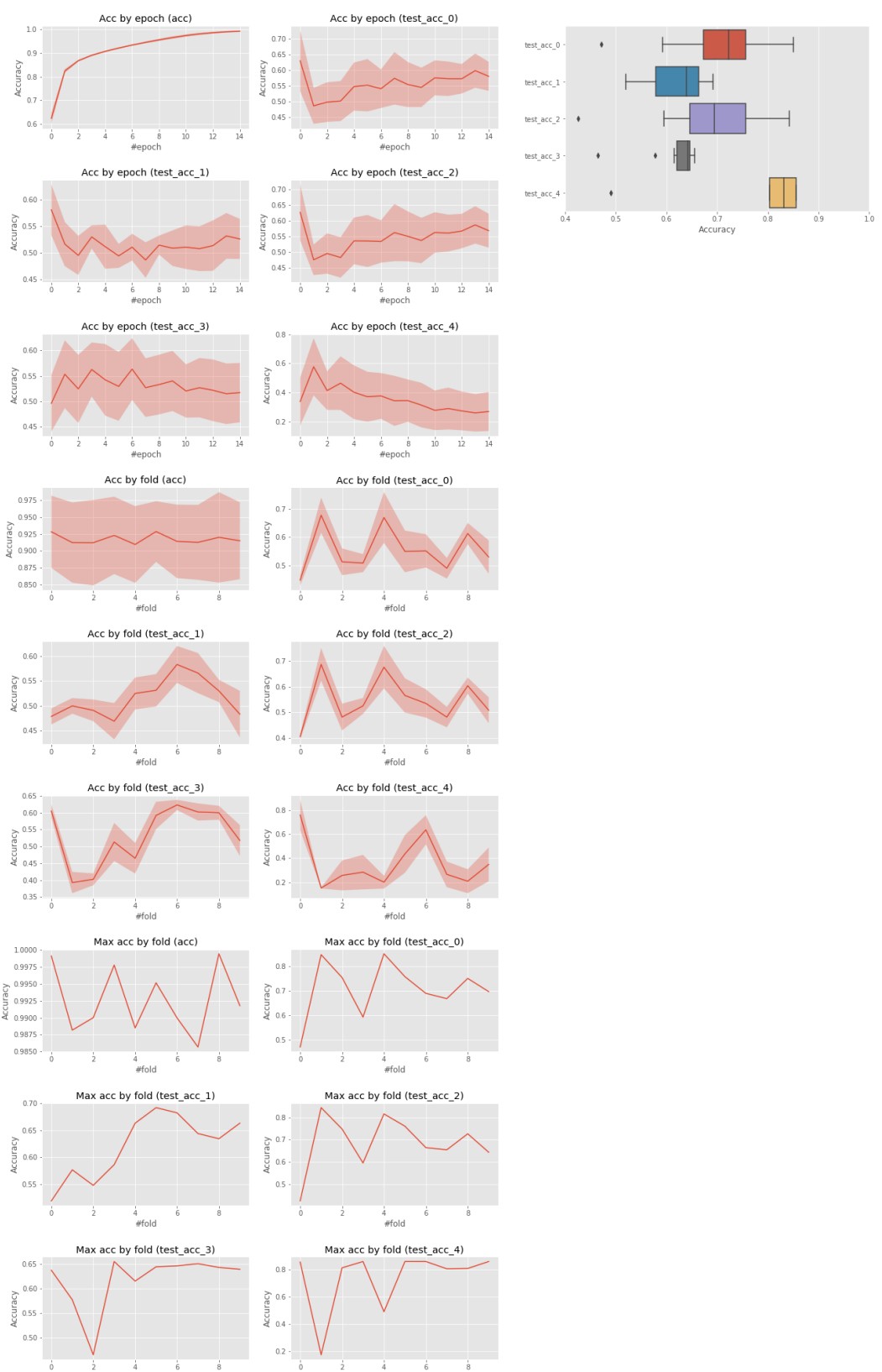

Figure 6: max_lr=0.005, lr_decay=0.75

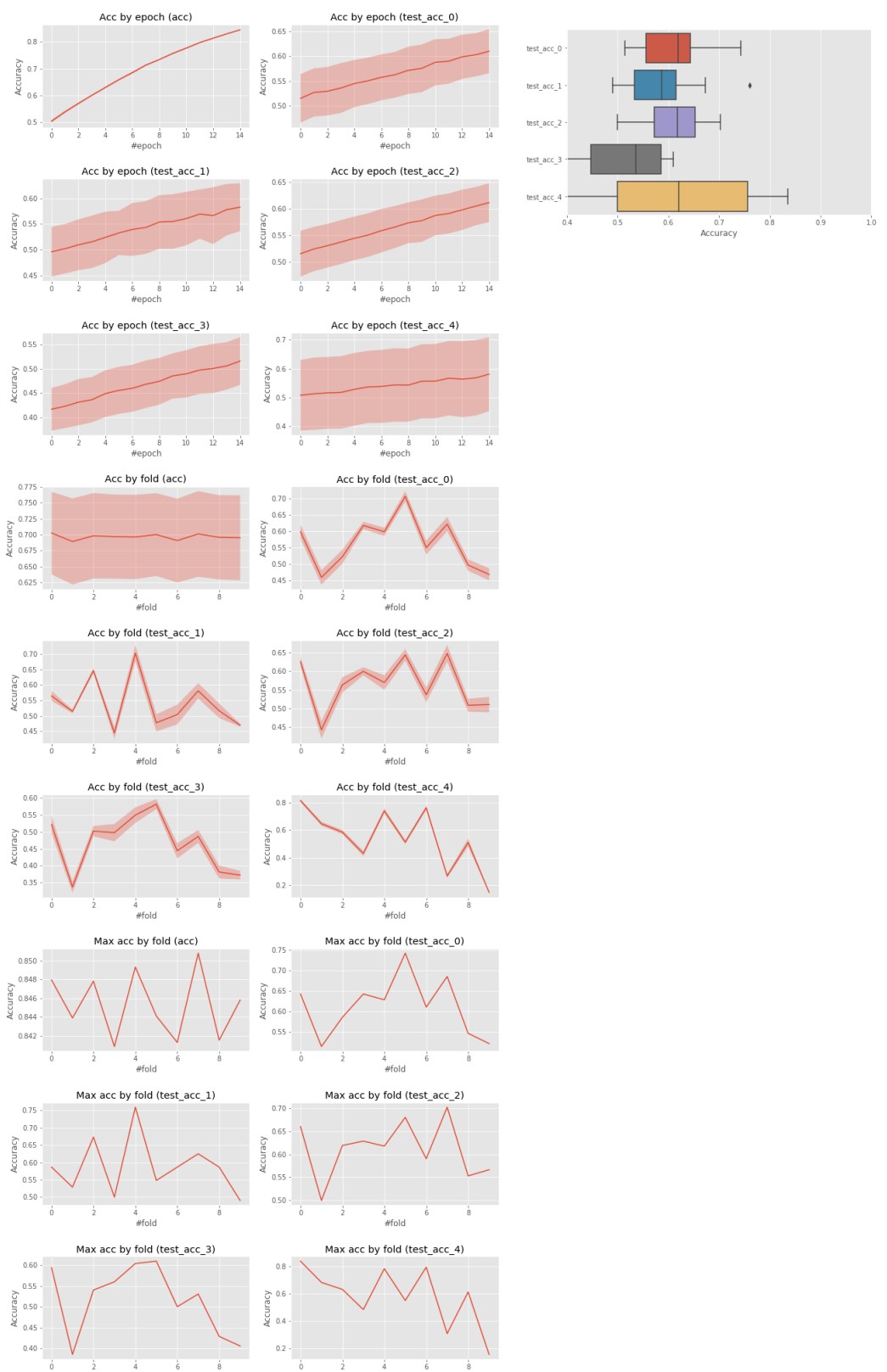

Figure 7: base_lr=0.0005, max_lr=0.001, cyclic_mode=exp_range

