# OpenReview forum: "Multi-Task Multicriteria Hyperparameter Optimization"
_ICLR.cc/2021/Conference — Reject_

### Official Review · AnonReviewer2 · 2020-10-24
**Review of Multi-Task Multicriteria Hyperparameter Optimization**

**Rating:** 3
**Confidence:** 4

**Review:**

This article bases hyperparameter optimization over multi-objective by aggregating them with weights. An illustration is provided on a grid search to select Pareto optimal solutions.

Aggregation of objectives to get scalar values has been studied at length in multi-objective optimization, so it is hardly novel. See, e.g.,  Miettinen, K., Nonlinear multiobjective optimization,  Springer, 1999, 12.
Besides , choosing weights is known to be quite difficult in practice.

Concerning optimization of hyperparameters, there are many works applicable that would be much more efficient than a crude grid search. In the case of Bayesian optimization, there are works like, e.g.,:
-Swersky, K.; Snoek, J. & Adams, R. P., Multi-task bayesian optimization, Advances in neural information processing systems, 2013, 2004-2012
- Paria, B., Kandasamy, K., & Póczos, B. (2020, August). A flexible framework for multi-objective Bayesian optimization using random scalarizations. In Uncertainty in Artificial Intelligence (pp. 766-776). PMLR.
- Hernández-Lobato, D., Hernandez-Lobato, J., Shah, A., & Adams, R. (2016, June). Predictive entropy search for multi-objective bayesian optimization. In International Conference on Machine Learning (pp. 1492-1501).

Since no comparison is provided with methods from the literature, that the proposed method is not novel, I thus recommend rejection.

Typos:
Eq. 4: sise → size
P2: the nearest Pareto front to the origin?

---

> ### Author Response · Authors · 2020-11-15
> **Added cited articles and fixed typos**
>
> Thanks to the reviewer for the disadvantages described in our review. We have added the articles provided by the reviewer to the literature of our article on page 2 in section 2. But these articles use either only multi criteria (multi-objective) or only multi-task optimization. Our method combines these two optimizations. We also thank the reviewer for the typos found. We have corrected them.
>
> In [1, 3, 4], methods of multi-object optimization are described, rather than a combination of multi-objective and multi-task optimization.
>
> Methods of multi-task optimization are described in [2]. But these methods also do not combine multi-objective and multi-task optimization.
>
> [1] Miettinen, K., Nonlinear multiobjective optimization, Springer, 1999.
>
> [2] Swersky, K.; Snoek, J. & Adams, R. P., Multi-task bayesian optimization, Advances in neural information processing systems, 2013, 2004-2012.
>
> [3] Paria, B., Kandasamy, K., & Póczos, B. (2020, August). A flexible framework for multi-objective Bayesian optimization using random scalarizations. In Uncertainty in Artificial Intelligence (pp. 766-776). PMLR.
>
> [4] Hernández-Lobato, D., Hernandez-Lobato, J., Shah, A., & Adams, R. (2016, June). Predictive entropy search for multi-objective bayesian optimization. In International Conference on Machine Learning (pp. 1492-1501).

---

> > ### Comment · AnonReviewer2 · 2020-11-16
> > **Response**
> >
> > Thanks to the authors for improving their paper and replying to comments.
> > I agree that a multi-task multi-objective Bayesian optimization algorithm may not be  directly available, but simple versions can be coded as the required components are complementary. It would provide pertinent methods to compare with, which is currently missing as noted also by other reviewers.

---

### Official Review · AnonReviewer1 · 2020-10-27
**clear reject**

**Rating:** 2
**Confidence:** 4

**Review:**


This paper proposes a multi-task multicriteria hyperparameter optimization method. Experiments fail to demonstrate the performance of the described method.

Strong points:

- The multi-objective selection procedure appears sensible
- Experiments run on state of the art models

Weak points:

- Experiments don't say what dataset they used, don't make me look at the citation when you have about 4 pages left of space
- Real values for objectives never provided
- No baseline / comparison
- No justification for method choice

Justification:

The paper makes little sense to me. A lot of things are poorly explained, although somewhere in there is the core of a good idea. The whole paper needs to be rewritten with an emphasis on properly defining terms, objectives and research questions.

I strongly recommend that this paper be rejected.

Some suggestions:
 - I am not sure if hyperparameter optimization is the right term to describe this method. One cannot speak of hyperparameter optimization, as there is no hyperparameter optimization process taking place. In fact, this is more akin to a selection procedure.
 - Equation 9 is introduced as the vector of the optimal solution, whereas it is one of many optimal solutions. It is incorrect to say that the optimal solution is putting an equal weight on all objectives, as far as multi-objective optimization goes.
 - Add some figures explaining the whole procedure (from hyperparameter evaluation to multi-objective selection)
 - Evaluate against baselines
 - Provide some justification for the steps of the method (i.e. in 3.2)
 - Have someone not familiar with the work revise the paper before submitting.

---

> ### Author Response · Authors · 2020-11-15
> **Added dataset description, added plots for the learning curve, added justification for method choice, added figures to explain the steps of our method and added justification for the steps of our method**
>
> We thank the reviewer for the constructive feedback! We have made changes in the paper to better address your concerns. Although there are few points of disagreement, we are definitely respectful and grateful to the reviewer for the valuable comments.
>
> **Experiments don't say what dataset they used** The dataset used consists of tasks. Each task consists of several images of one of two objects such as a plastic bottle and other object. The tasks differ in how the images are made, namely lighting, background and used cameras. Please see page 4 section 4.1 for the change.
>
> **Real values for objectives never provided** We have added plots for the learning curve of the neural network for each test set and each epoch for the obtained optimal hyperparameters. Please see Appendix C for the change.
>
> **No baseline / comparison. Evaluate against baselines** We agree that it is a significant drawback of our work. In future research, we will eliminate it and do more extensive experiments.
>
> **No justification for method choice** Pareto filters out obviously non-optimal solutions among all combinations of hyperparameters and keeps those that are optimal according to some criteria. Using the criteria, we choose among the Pareto optimal solutions those that are optimal according to the given criterion. That is, we put a correspondence between the Pareto optimal solutions and criteria. In the future, if we want to choose solutions according to other new criteria, then it is enough to calculate the objective function for the new criteria and choose among the previously obtained Pareto solutions that are optimal for these new criteria. Thus, we spend an excessive amount of time training neural networks with all combinations of hyperparameters, but we save time when evaluating. Please see page 2 section 2 for changes.
>
> **Hyperparameter optimization or a selection procedure** In [1] it is said that hyperparameter optimization is the choice of such hyperparameters at which the objective function reaches a minimum. This is the same as the optimization problem described in our article (equation (1)). Therefore, we say that in our article hyperparameter optimization is carried out, and not a selection procedure.
>
> **Equal weight for all objectives is not an optimal solution** We agree that such a solution is not optimal in terms of the objective function. But here it is not about the objective function, it is about the relation between the criteria. That is, in this case, the optimal means not to give preference to any criterion, but to equalize them with each other and choose a solution that corresponds these equal criteria. Thus, we say that equal weight is correct for choosing the optimal solution.
>
> **Add some figures explaining the whole procedure** We have added six figures to explain the steps of our method. Please see page 3 section 3.2 for the change.
>
> **Provide some justification for the steps of the method** Pareto gives solutions that are closer to optimal by some criterion, and farther from optimal by other criteria. These solutions can already be called optimal. But they are optimal according to various criteria. To determine by what criteria these Pareto-selected solutions are optimal, we calculate the objective function for different criteria. For these specified criteria, we obtain the corresponding optimal solutions. See changed section 3.2.
>
> **Have someone not familiar with the work revise the paper before submitting** Thanks for the suggestion. We will take it into account in our future research.
>
> **The whole paper needs to be rewritten with an emphasis on properly defining terms, objectives and research questions** We have added a definition of task and criterion, and research objective. We agree with the reviewer that the article needs to be significantly improved.
>
> [1] James Bergstra and Yoshua Bengio. Random search for hyper-parameter optimization. Journal of Machine Learning Research, 13(Feb):281–305, 2012.

---

> > ### Comment · AnonReviewer1 · 2020-11-20
> > **response**
> >
> > I will start by thanking the authors for their rebuttal. I have read it and I must say that sadly it does not change my opinion of the paper. This is still very much a preliminary work and the quality is not high enough to be published at a conference like ICLR. Best of luck!

---

> > > ### Author Response · Authors · 2020-11-21
> > > **Response**
> > >
> > > Tell us, please, in your opinion, this article is preliminary because there are no comparisons with baselines?

---

> > > > ### Comment · AnonReviewer1 · 2020-11-24
> > > > **suggestions**
> > > >
> > > > Some pointers:
> > > >
> > > > - The introduction is very short and could be improved to provide some justification as to why the multi-task multiobjective problem is important, or an example of situation where this problem arises.
> > > >
> > > > - Equation 6 is not presented by text
> > > >
> > > > - Section 3 overall is still hard to follow and should be improved. See some points which drew my attention :
> > > > -- "1. The vectors from the evaluation V (the number of such vectors is N_criteria) is in the space of given crit...", I think you mean the number of such vectors is N_combinations?
> > > > -- "2. Pareto gives solutions that are closer to optimal by some criterion, and farther from optimal by
> > > > other criteria", -- Rephrase "closer to optimal" and "farther from optimal". A point is either Pareto optimal or not. Maybe formalize the concept of Pareto optimality. Google might come in handy here as this is something that is often defined.
> > > > -- "nearest Pareto Front" I think by definition the Pareto Front is the set of Pareto optimal solutions and therefore is there is no "nearest Pareto Front to the origin"
> > > > -- equation 8 contains a mistake: $... \upsilon_{min} \in R^{N_{criteria}}, \upsilon_{min} \in R^{N_{criteria}}$ should be $\upsilon_{min} \in R^{N_{criteria}}, \upsilon_{max} \in R^{N_{criteria}}$
> > > > -- "Thus, the optimal solution is the solution closest to the origin of coordinates", I still disagree on this point. Optimal according to what distance metric? to what weighting of the objectives? you should probably explicitly say that it would be optimal with an equal weighting of objectives, which would at least be unambiguous.
> > > >
> > > > - It is not clear to me how your method handles the multiple tasks... the task i is not mentioned anymore after equation 6 in section 3. Is the performance over the different tasks averaged? what treatment is applied? Are tasks simply concatenated in the space of criteria? Also define the tasks explicitly in Section 4, give examples.
> > > >
> > > > - However, the main flaw remains indeed in the lack of comparisons. I am positive that there are many methods out there to compare with, even if they are only naive approaches (such as the one suggested by reviewer 2). Without baselines, we are left wondering if this is improving over anything. If there no simple baseline to evaluate against, then you must convince the reader why this is a hard problem, a thing the paper does not do in its current form.

---

> > > > > ### Author Response · Authors · 2020-11-24
> > > > > **Response**
> > > > >
> > > > > Thank you for these suggestions! We will make changes and improve our further research.

---

### Official Review · AnonReviewer3 · 2020-10-28
**Half-baked, needs more work**

**Rating:** 3
**Confidence:** 4

**Review:**

This paper proposes MTMC, a method that solves the pareto optimization problem for multiple tasks and multiple criteria.

Pro:
1, concise.
2, solution seems interesting from a technical perspective.

Con:
1, No explanation on why this proposed method should work well.
2, No comparison to state-of-the-art method (or any baseline methods). Several related methods were discussed in related work, but why not compare to them, e.g. (Igel, 2005)?
3, Experiments are not convincing. Please include descriptions of this problem in (Akhmetzyanov & Yuzhakov, 2019) and add more realistic real-world problems. Please include specific description of the metric to evaluate different methods including the proposed one.

Some minor points:
1, It's unclear what N_combination is exactly and why it's different from N_parameter.
2, Typo: in Eq (4), x_sise -> x_size

Overall I think this paper is half developed and can benefit from some intuitive explanations, theoretical analysis of the proposed method and more convincing experiment.

---

> ### Author Response · Authors · 2020-11-15
> **Added plots for the learning curve, added a description of the problem, added cited articles and fixed typo**
>
> We thank the reviewer for the helpful comments! We are glad to hear that our solution is interesting from a technical point of view.
>
> **Why this proposed method should work well** In our method, Pareto optimality means that it is impossible to improve the Pareto optimal solution by any criteria without worsening it by at least one other criteria. Thus, for a certain set of values ​​of the criteria, the Pareto-selected solutions are optimal. These optimal solutions corresponds to some criteria. We set the criteria. Further, from these Pareto optimal solutions, it is necessary to determine the closest criteria to the given criteria. In our method, this is done by finding the minimum weighted sum of Pareto solutions (where the weight is the inverse criteria). Please see page 2 for change.
>
> **No comparison to state-of-the-art method** We agree that no comparison with baselines is a significant drawback of our work. In future research, we will eliminate it and do more extensive experiments.
>
> **Experiments are not convincing** We have added plots for the learning curve of the neural network for each test set and each epoch for the obtained optimal hyperparameters. Please see Appendix C for the change.
>
> **Include descriptions of this problem** Yes, we should have added a description of the problem. Let's first introduce the term task. *Task* means a set of images with a number of classes $ N_{classes} $ and a number of images $ N_{images} $. There are examples of the same classes between tasks, the difference is how the images are made (different lighting, background and used cameras). The problem is in the choice of such hyperparameters, which achieve the highest classification accuracy among several tasks. Each task consists of several images of one of two objects such as a plastic bottle and other object. The tasks differ in how the images are made, namely lighting, background and used cameras. Please see page 4 section 4.1 for changes.
>
> **Add more realistic real-world problems** We have added cited articles for computer vision, robotics, natural language processing and speech synthesis on page 1 in section 2.
>
> **What N_combination is exactly and why it's different from N_parameter** There is hyperparameter set $ H $. $ N_{parameter} $ is size of the set $ H $, $ N_{parameter} = |H| $. Some hyperparameters are quantitative (for example, learning rate), other ones are categorical (for example, learning method). Quantitative hyperparameters are in a certain interval with a fixed step. Search in the proposed method is performed on all combinations of these hyperparameter values, and the number of such combinations is indicated by $ N_{combination} $.
>
> **Typo: in Eq (4), x_sise -> x_size** Thanks for the typo found. We fixed it.

---

### Official Review · AnonReviewer4 · 2020-10-29
**Relevant topic, paper appears a bit preliminary**

**Rating:** 2
**Confidence:** 4

**Review:**

Summary
The paper addresses the proposes a solution for multi-task multicriteria hyperparameter optimization. The solution hinges around Pareto optimality. The technique appears to do an exhaustive enumeration in the space of hyperparameters (Equation 4). Experimental results are provided for two scenarios with two and three hyperparameters.

Pros
The paper addresses an important problem.

Cons
1.	The paper appears to use exhaustive enumeration before finding Pateto frontier. Exhaustive enumeration is computationally expensive.
2.	The experimental results are sketchy. The final performance of the models as a result of hyper-parameter selection is not provided.
3.	Comparison with baselines is missing.

Clarifications needed
The paper can be improved by adding the following information.

1.	Section 4.1: Provide details of the tasks.
2.	Section 4.1: “Neural networks trained on ten TPUs v2, which took several days.”: How many days?
3.	Section 4.2: “sample mean/variance of the epoch number at which convergence is achieved in the test sample.”: What is meant by convergence on test samples?

---

> ### Author Response · Authors · 2020-11-15
> **Added plots for the learning curve and added a task definition**
>
> We want to thank the reviewer for helpful comments. We have made changes in the paper to make it clearer and to better address your concerns.
>
> **Exhaustive enumeration is computationally expensive** Yes, we agree with that. But on the other hand, MTMC gives hyperparameters for various criterion weights without additional training of the neural network. That is, when training the neural network with all combinations of hyperparameters, an excessive amount of time is spent, but when evaluating the neural network, we save time.
>
> **Performance is not provided** We have added plots for the learning curve of the neural network for each test set and each epoch for the obtained optimal hyperparameters. Please see Appendix C for the change.
>
> **Comparison with baselines is missing** We agree that it is a significant drawback of our work. In future research, we will eliminate it and do more extensive experiments.
>
> **Provide details of the tasks** In our article, a task means a set of images with a number of classes $ N_{classes} $ and a number of images $ N_{images} $. There are examples of the same classes between tasks, the difference is how the images are made (different lighting, background and used cameras). On page 1 in section 2 we have added a task definition.
>
> **Number of training days** Unfortunately, now it is difficult to say exactly how many days the training took, since the experiments were carried out a long time ago.
>
> **Convergence on test samples** Here we mean the achievement of a small change in accuracy of the neural network on a test set after training for several epochs. If we denote $ \epsilon_{i} = accuracy_{i}-accuracy_{i-1} $, then we can say that the neural network converged on the test set at epoch $ i $, when $ \epsilon_{i} $ has a small value.

---

### Decision · Program_Chairs · 2021-01-07
**Final Decision**

**Decision:**

Reject

**Comment:**

The paper has been discussed by the reviewers that have acknowledged the rebuttal and the authors’ responses. However, the reviewers still had the following weaknesses and concerns (not solved post rebuttal):

* Expensive procedure (e.g., exhaustive enumeration before finding Pareto frontier)
* The experiments should be more rigorous, with more realistic real-world problems.
* Missing comparison with baselines (unanimously acknowledged by the reviewers).
* No explanations and insights provided as to why the method should work well
* Clarity of the presentation

As a result, the paper is recommended for rejection. The detailed comments of the reviewers provide an actionable list of items to improve the paper for a future resubmission.